# Auto Aligning, Error-Compensated Broadband Collimated Transmission Spectroscopy

**DOI:** 10.3390/s24216993

**Published:** 2024-10-30

**Authors:** Karsten Pink, Alwin Kienle, Florian Foschum

**Affiliations:** 1Institut für Lasertechnologien in der Medizin und Meßtechnik an der Universität Ulm, 89081 Ulm, Germany; alwin.kienle@ilm-ulm.de (A.K.); florian.foschum@ilm-ulm.de (F.F.); 2Department of Physics, Ulm University, 89069 Ulm, Germany

**Keywords:** collimated transmission, spectroscopy, wide spectral range, Monte Carlo simulation, polystyrene spheres, Mie calculation

## Abstract

Broadband spectral measurements of the ballistic transmission of scattering samples are challenging. The presented work shows an approach that includes a broadband system and an automated adjustment unit for compensation of angular distortions caused by non-plane-parallel samples. The limits of the system in terms of optimal transmission and detected forward scattering influenced by the scattering phase function are investigated. We built and validated a setup that measures the collimated transmission signal in a spectral range from 300 nm to 2150 nm. The system was validated using polystyrene spheres and Mie calculations. The limits of the system in terms of optimal transmission and detected forward scattering were researched. The optimal working parameters of the system, analyzed by simulations using the Monte Carlo method, show that the transmission should be larger than 10% and less than 90% to allow for a reliable measurement with acceptable errors caused by noise and systematic errors of the system. The optimal transmission range is between 25% and 50%. We show that the phase function is important when considering the accuracy of the measurement. For strongly forward-scattering samples, errors of up to 80% can be observed, even for a very small numerical aperture of $6.6\cdot 10^{-4}$, as used in our experimental system. We also show that errors increase with optical thickness as the ballistic transmission decreases and the multiscattered fraction increases. In addition, errors caused by multiple reflections in the sample layer were analyzed and also classified as relevant for classical absorption spectroscopy.

## 1. Introduction

Collimated transmission describes the unscattered and unabsorbed light passing through a sample. By using collimated transmission spectroscopy, the extinction coefficient can be determined. In general, liquid samples, as well as solid samples, are investigated using a collimated transmission-based measurement system. The spectrally resolved optical properties carry a lot of information about the physical and chemical behavior of a researched object. If either the absorption coefficient or the scattering coefficient is known, one can identify the other parameter by measuring the extinction coefficient. Collimated transmission and absorption spectroscopy have been used in various publications and existing measurement systems. These methods are used to analyze food and food substitutes, especially in liquid and emulsion forms [1,2,3,4,5,6], as well as more technical emulsions, like drilling waters or cooling lubricants [7,8,9]. For some applications, a monochromatic measurement is sufficient. For example, a monochromatic measurement is suitable for determining the concentration of a chromophore in a watery solution using the Lambert–Beer law [10,11]. Collimated transmission, as a spectroscopic system, is further used to determine the extinction coefficient of teeth. This is supposed to help develop restoration materials that fulfill aesthetic requirements [12,13]. Special polarized applications are used to determine the extinction coefficient of tissue like phantoms [14]. Additionally, collimated transmission spectroscopy can be used to determine, together with further measurements like integrating sphere measurements, the anisotropy factor of various materials. Collimated transmission measurements can be compared to theoretical data, for example, using Mie theory for scattering phantoms [15,16].

In this work, a technical setup to acquire a broadband collimated transmission spectrum is presented for the first time, to the best of our knowledge. This setup takes into account, chromatic aberrations and etaloning and fiber properties, as well as imperfections in components and the applied scattering phase function. The impact of these errors is shown, and our solutions are described and discussed in this paper. The change in path length due to multireflection and the errors caused by neglecting path length when evaluating detected light are demonstrated. Furthermore, errors in measurements of the collimated transmission spectrum due to multiscattering and forward scattering are discussed and classified in terms of the degree of transmission and scattering phase functions of the samples.

## 2. Materials and Methods

The experimental setup, including the evaluation of the results, the control of the components, the theoretical background, and the type of simulations carried out in this work are described in the following subsections.

### 2.1. Experimental Setup

The experimental setup is dividable into three main units: the illumination unit, which includes an automatic xy adjustment stage; beam propagation through the sample section; and the detection unit. In Figure 1, a schematic sketch of the developed setup is presented. A laser-pumped xenon plasma lamp (XWS-30, ISTEQ, High Tech Campus 9, 5656 AE Eindhoven, The Netherlands) is coupled via a 400 µm broadband solarized fiber (Avantes B.V., Oude Apeldoornseweg 28, 7333 NS Apeldoorn, The Netherlands) to the xy stage (XY-S) (MTOM2-1,Vision GmbH, Weg der Vision 1, 30890 Barsinghausen, Germany), which serves as a light source (LS). The unit includes a pump laser, a xenon lamp, and a complete control unit. By passing through a pinhole (PH) with a diameter of 100 µm, good collimation is enabled by a parabolic off-axis mirror (POM) (MPD169-F01-Ø1” 90° Off-Axis Parabolic Mirror, Thorlabs Inc., Newton, NJ, USA) with a focal length of f = 152 mm and a resulting numerical aperture (NA) of $3.29\cdot 10^{-4}$. The collimated light propagates through the shutter (S) and onto the sample (SA). The remaining ballistic fraction of the light is focused by a second identical parabolic off-axis mirror (f = 152 mm) onto a pinhole with a diameter of 200 µm, which results in an NA of $6.58\cdot 10^{-4}$. Directly after the pinhole, a special broadband optical fiber (F) (Avantes B.V., Oude Apeldoornseweg 28, 7333 NS Apeldoorn, The Netherlands) with a core diameter of 600 µm is located. Via a customized broadband y fiber (Y-F) (Avantes B.V., Oude Apeldoornseweg 28, 7333 NS Apeldoorn, The Netherlands) with a diameter of 400 µm, followed by two legs with diameters of 200 µm, the light is then coupled into two spectrometers (ultraviolet-visible and near-infrared spectrometers; UV-VIS-S and NIR-S, respectively) covering spectral ranges of 250 nm to 1100 nm (MAYA SC Pro, ocean insight, Orlando, FL, USA) and 900 nm to 2150 nm (BTS2048-IR, Gigahertz Optik GmbH, An der Kälberweide 12, 82299 Türkenfeld, Germany), respectively. In the setup, only mirrors are used to avoid chromatic aberrations. The illumination fiber and the detection fiber in front of the Y fiber are moved continuously during the measurements to avoid intensity shifts due to fiber modes, using a custom-made vibration device. The detection fiber has a length of 10 m to obtain an almost homogeneous exit light. This is very important to keep the overlaying etaloning signal constant for each wavelength. All the components can be controlled via a personal computer (PC). The xy stage is able to move the fiber end connected to the light source in the focal plane of the POM using a feedback loop to automatically find the position with the strongest detected signal. This is necessary to handle imperfectlyparallel sample boundaries. Such samples add an angle to the ballistic light, which causes a spacial shift in the focal plane of the detection POM. By moving the illumination fiber and, therefore, adding an angle relative to the optical axis of the system to the illumination beam, this effect can be compensated for. The used algorithm is described in Section 2.4.

### 2.2. Samples for Validation

To validate the setup, very well characterized polystyrene microspheres (microParticles GmbH, Volmerstr. 9A, H3.5.1, D-12489 Berlin, Germany) with two different diameters of 3.97 µm (nominal) and 3.917 µm (measured), with a measured standard deviation of 0.03 µm, and 2.0 µm (nominal) and 2.002 µm (measured), with a measured standard deviation of 0.012 µm, were used. The deviations from the specified nominal diameter are the result of the best fit of the parameters to the measurements using Mie theory. These deviations may be caused by the variability in the manufacturing process or by the manufacturer’s sizing method. The absorption of these samples is negligible, which means the absorption coefficient is $\mu_{a}=0$ in the considered wavelength range. Therefore, the extinction coefficient $\mu_{t}$ is equal to the scattering coefficient $\mu_{s}$ because of their relationship $\mu_{t}=\mu_{a}+\mu_{s}$. Demineralized water was used to suspend the particles for the measurements. Additionally, the concentration of the spheres was sufficiently low to avoid dependant scattering. The measurements were compared to calculations of the spectrally resolved scattering coefficient based on Mie theory, which provides the exact solution. Standard cuvettes (Typ 101-QS 10 × 10 mm, Hellma GmbH & Co., KG, Klosterrunsstraße 5, 79379 Muellheim, Germany) with a thickness of 10 mm and customized cuvettes with a thickness of 0.5 mm were used for the measurements. Both cuvettes use the same quartz glass (QS, Präzisions Glas und Optik GmbH, Im Langen Busch 14, 58640 Iserlohn, Germany). The corresponding refractive index of quartz glass was used for calculations with cuvettes and to correct the measurements.

### 2.3. Multiple Reflections in Plane-Parallel Samples

Due to the refractive index mismatch between the sample and the surrounding, the light is also affected by boundary reflections. Light which was reflected multiple times can also enter the detector, but has a substantially longer path within the sample material [17]. In Figure 2, a schematic representation illustrates this behavior. If the longer path lengths are not taken into to account in the Lambert–Beers law evaluation, errors occur even for non-scattering media, which is also relevant for standard absorption spectroscopy. To overcome this issue, the total transmission can be calculated by taking into account the transmission through the medium and the reflections at the boundaries. A geometric series depending on the sample structure can be determined. For different refractive indices of the sample ($n_{2}$) and the surrounding medium ($n_{1}$), multiple reflections have to be considered. For samples composed of multiple slabs, such as a cuvette, all of the interfaces must be taken into account.

To determine the total transmission, the different light paths that are detected can be defined separately. $T_{I}$ represents the transmission through a single interface layer and can be calculated using the transmission formulas for perpendicular incidents $T_{I1}=T_{I2}=T_{I}$ for a single slab in a random medium using Fresnel’s formula. This is represented in Equations (Equation 1) and (Equation 2):(1)$$\begin{array}{c} T_{I1}=1-{\left(\frac{n_{1}-n_{2}}{n_{1}+n_{2}}\right)}^{2} \end{array}$$
(2)$$\begin{array}{c} T_{I2}=1-{\left(\frac{n_{2}-n_{1}}{n_{2}+n_{1}}\right)}^{2}. \end{array}$$

The transmission through the scattering and absorbing medium is given by $T_{M}$ and can be calculated by using Lambert–Beer’s law. The *R* values describe the reflected light after a defined number of reflection and transmission processes within the sample slab. Similarly, the *T* values stand for the transmitted light. The individual $R_{n}$ and $T_{n}$ values can be calculated by
(3)$$\begin{array}{cc} R_{F} & =1-T_{I}\\ R_{0} & =T_{I}^{2}T_{M}^{2}\left(1-T_{I}\right)\\ R_{1} & =T_{I}^{2}T_{M}^{4}{\left(1-T_{I}\right)}^{3}\\ R_{2} & =T_{I}^{2}T_{M}^{6}{\left(1-T_{I}\right)}^{5}\\ \; & \ldots \ldots \end{array}$$
(4)$$\begin{array}{cc} T_{0} & =T_{I}^{2}T_{M}\\ T_{1} & =T_{I}^{2}T_{M}^{3}{\left(1-T_{I}\right)}^{2}\\ T_{2} & =T_{I}^{2}T_{M}^{5}{\left(1-T_{I}\right)}^{4}\\ T_{3} & =T_{I}^{2}T_{M}^{7}{\left(1-T_{I}\right)}^{6}\\ \; & \ldots \ldots \end{array}$$

The total *R* and *T*, using the transmission values of the sample, can be calculated by
(5)$$R_{tot}=R_{F}+T_{I}^{2}\sum_{j=0}^{\infty }T_{M}^{2j+2}{\left(1-T_{I}\right)}^{2j+1}$$
and
(6)$$T_{tot}=T_{I}^{2}\sum_{j=0}^{\infty }T_{M}^{2j+1}{\left(1-T_{I}\right)}^{2j}.$$

These sums converge to a geometric series which leads to
(7)$$R_{tot}=\left(1-T_{I}\right)+\frac{\left(1-T_{I}\right)T_{I}^{2}T_{M}^{2}}{1-T_{M}^{2}\left(1-T_{I}^{2}\right)}$$
and
(8)$$T_{tot}=\frac{T_{I}^{2}T_{M}}{1-T_{M}^{2}{\left(1-T_{I}\right)}^{2}}.$$

Using the presented equations, the transmission of the medium can be obtained with
(9)$$T_{M}=\frac{-T_{I}^{2}\pm sqrt\left(T_{I}^{4}+4T_{tot}^{2}{\left(1-T_{I}\right)}^{2}\right)}{2T_{tot}{\left(1-T_{I}\right)}^{2}}.$$

If a cuvette is used, the presented calculations can be modified. Figure 3 schematically shows the different slabs of a cuvette. The transmission through the respective glass slab is $T_{G}$.

For non-absorbing and non-scattering glass, $T_{G}$ can be calculated by expanding Equation (Equation 8) for two different boundaries. $T_{I1}$ and $T_{I2}$ can be substituted by $T_{G1}$ and $T_{G2}$ and can be adjusted to the interface transmission between the glass and medium and between the glass and air. This leads to
(10)$$T_{G}=\frac{T_{G1}T_{G2}}{1-\left(1-T_{G1}\right)\left(1-T_{G2}\right)}.$$

$T_{M}$ can now be determined using Equation (Equation 9) and replacing $T_{I}$ with $T_{G}$.

### 2.4. Automated Self-Centering Algorithm

Non-parallel samples result in an additional angle to the collimated beam, which can reduce the detected intensity. To compensate for this effect, the designed system is able to move the light source fiber, including the pinhole, in the focal plane of the parabolic off-axis mirror on the illumination side. This allows an angle to be applied to the collimated beam. This effect is used to compensate for the additional angle experienced by ballistic light from non-plane-parallel diffracting samples. The system uses a feedback loop to automatically find the position for the angle of incidence with the strongest signal by evaluating the integral over all wavelengths. The algorithm for determining the optimum angle of incidence starts from a predefined position on the optical axis of the system. From there, if no intensity above a certain threshold is found, the position of the light source is modified in a circular manner, keeping the distances between individual measurements within a defined range so that the signal cannot be missed. A schematic representation is shown in Figure 4. The distance between measurements was determined by the diameter of the fiber core and the focal length of the detection POM of the system. The red circles represent the area around the position of the light source where a signal can be detected. Once a signal above a certain threshold has been detected, an optimization algorithm uses the two dimensional gradient to search for the optimum position. To improve this algorithm, the step sizes between measurement positions are dynamically changed as a function of the gradient. If a sufficient signal is not found at any point, the measurement is cancelled.

### 2.5. Monte Carlo Method Used for Verification

In this work, a standard Monte Carlo method (MC method) was used to solve the radiative transfer equation (RTE). The components used in the experiment and all parameters were digitally modeled in this simulation. The MC method used is a version of a simulation described in [15] and was developed in house. The major advantage of this simulation is the possibility to include all optical components in the simulation in order to reproduce the most realistic behavior of the system possible. The method uses the scattering coefficient, the absorption coefficient, the scattering phase function, and the refractive index to simulate energy packages propagating through a system. The simulation recovers the exact solution of the RTE for an infinite number of calculated photons. The used Monte Carlo method did not consider the polarization state of the propagated light. This was not necessary because the light source used was unpolarized. Neglecting the polarization in the simulation improves the computational speed by several orders of magnitude.

## 3. Results

This paragraph shows the results of various simulations as well as the experimental results.

### 3.1. Errors Caused by Multireflection

Section 2.3 describes the method for determining the transmission through the medium including multireflection. To investigate the extent of the error caused by neglecting these reflections, simulations were carried out. The comparison of the calculated collimated transmission values, using the modified version of Equation (Equation 9), with the described MC simulation show perfect agreement within the statistical uncertainties. In order to reproduce the standard procedure, which does not consider the light reflection within the cuvette, used for absorption spectroscopy, forward calculations of a non-scattering but absorbing medium (‘sample’ measurements) were referenced to forward calculations using a non-absorbing non-scattering medium in the cuvette (‘reference’ measurement). The resulting transmission values were evaluated for the absorption coefficient using Lambert–Beer’s law $\mu_{t}=-\frac{1}{cd}\;ln\left(\frac{I}{I_{0}}\right)$, where $\mu_{t}$ is the extinction coefficient, *I* is the measured intensity, $I_{0}$ is the reference intensity without the sample, *c* represents the concentration, and *d* is the sample thickness. This was compared with calculations taking into account multireflection and the resulting change in the effective light path. The relative error between the resulting absorption coefficients of the different evaluations is shown in Figure 5. Different refractive indices were tested for the glass of the cuvette and the measured medium. As expected, the larger the refractive index mismatch between the glass and sample, the larger the relative error. The cuvette size was 10 mm. The refractive indices for the second calculation shown (wide dashed line) were chosen to fit a standard glass cuvette filled with a water-based sample. The error is larger for smaller absorption coefficients and converges toward a limited value for decreasing absorption, depending on the refractive index relation. The error converges toward zero for increasing absorption due to the small amount of reflected light.

These calculations show that when using the standard method for commonly used cuvettes, an error of more than 0.5% will occur. If the refractive index ratio is increased, for example by using different cuvette glasses, the error can exceed 1%. In order to avoid these errors, the multireflection theory should be used in the analysis. For the presented method, all refractive indices must be known. These values can often be found in the literature. The refractive index of the sample can also be eliminated from the equation by making a reference measurement and then using the exact calculation for two different optical thicknesses.

### 3.2. Error Due to Forward Scattering

Scattering media can distort the collimated transmission measurement because the forward-scattered light contributes to the measured signal. This effect is highly dependent on the scattering characteristics of the sample. A variety of scatterers with different phase functions have been investigated to evaluate the error caused by forward scattering. For a general overview, the Heneye–Greenstein scattering phase function with different anisotropy factors $g=2\pi \int_{0}^{\pi }p\left(\theta \right)cos\left(\theta \right)sin\left(\theta \right)d\theta$ ranging from 0 to 0.999 was used. *p* is the normalized scattering phase function and θ is the scattering angle. Additionally, the results for scattering phase functions of microspheres obtained by Mie’s theory with a size distribution of the scatterers following a standard deviation of 0.1% of the different main particle sizes are presented. These are typical values for commercially available microspheres. The results are based on polystyrene spheres in aqueous suspension with refractive index values of 1.33 for water [18] and 1.586 for polystyrene [19]. Sphere diameters of 0.1 µm, 1 µm, 10 µm, and 100 µm were used to obtain a wide distribution of scattering phase functions. All calculated samples had a thickness of d_*s*_ = 10 mm and absorption was neglected.

Figure 6 shows the relative errors of the scattering coefficient µ_*s*_ for different optical thicknesses $\tau =\mu_{t}d_{s}$ using the Henyey Greenstein scattering phase function. The error is the difference between the µ_*s*_ used for the simulation and the scattering coefficient calculated by the intensity detected compared to a reference using Lambert–Beer’s law. The results vary significantly depending on the anisotropy factor. The colors indicate the different anisotropy factors. It is shown that for anisotropy factors lower than 0.99 and τ≈< 6, the error is sufficiently small. For an anisotropy factor of 0.99 and τ≈> 4, the error starts to increase drastically. For simulations with anisotropy factors of 0.99 and higher, an error is observed over the whole scattering coefficient range. The large errors are caused by the very large forward scattering. For τ≈> 7, the error increases. This is due to multiple scattering.

Figure 7 shows the relative errors of the scattering coefficient $\mu_{s}$ for different optical thicknesses τ using a scattering phase function calculated with Mie theory for different scatterer sizes as the input to the simulation and the scattering coefficient evaluated using the Lambert–Beer law. The colors indicate the different sizes of the scatterers., The first diffraction peak, and therefore the forward scattering in a narrow angle, is strongly correlated with particle size. For particle sizes of $d_{p}$ <= 10 µm and an optical thickness of τ≈< 10 the absolute values of the relative errors due to forward scattering are significantly below 1%. At an optical thickness of τ≈> 10, the absolute value of the relative error increases dramatically. For scatterers with a diameter of $d_{p}$ = 100 µm, errors of more than 15% were obtained for all optical thicknesses. The error starts to increase slowly at τ≈ 1 and the gradient steepens drastically at τ≈> 11. These effects appear due to the increase in the detected multiscattered light and the decrease in the detected ballistic light.

In Figure 8, the scattering phase functions for the different used scatterers are shown. Comparing the phase functions and keeping in mind the definition of the anisotropy factor, it becomes clear that the scattering of the first diffraction peak in a narrow angle has little effect on the anisotropy. Therefore, a higher anisotropy factor does not necessarily result in a larger error of the scattering coefficient. Instead, a large first diffraction peak leads to large errors.

### 3.3. Examinations to Determine the Optimal Transmission Range

To determine a transmission range where the setup achieves the most reliable results, further calculations were performed. A transmission signal was simulated using the multireflection theory. Relative errors were then added to the transmission and the relative error in the extinction coefficient was analyzed using Lambert–Beer theory including multireflection correction. Figure 9 shows the relative error depending on the transmission $t=I/I_{0}$ through the sample. The error added to the simulated signal is shown by the different curves and line styles. All of these results show an agreement toward a range of transmission rates where the error has the least effect. This range is identified between 25% and 50% transmission. Furthermore, it is noted that below 10% transmission and above 85% transmission, the error increases rapidly. The low transmission is of particular interest for biological samples, as these are often optically dense materials and the ability to produce thin samples is limited. On the other hand, transparent materials would need a large thickness to meet the requirements for optimal transmission. This could cause problems in manufacturing and sample holding.

### 3.4. Experimental Validation of the System

To validate the system, the collimated transmission through polystyrene spheres in various concentrations was measured. The monodispersity of the samples is very narrow, so oscillations in the extinction coefficient can be observed. These oscillations can be reproduced by Mie theory calculations. The measurements were made using a cuvette filled with demineralized water as a reference and the multireflection evaluation algorithm was used. Figure 10 shows the measurements in a standard cuvette with $d_{cuv1}$ = 10 mm, as described in Section 2.2. The samples were a suspension of polystyrene microspheres in demineralized water. The shown results display good agreement between the measurements and Mie calculations. Due to the uncertainty of the exact volume concentration of the scatterers in the initial suspension, the resulting scatterer concentration was manually adjusted in the Mie calculation. The microspheres have nominal diameters of 2.0 µm and 3.97 µm and are already described in Section 2.2. The calculations show a slightly different scatterer size of 2.002 µm and 3.917 µm to achieve the best fit, as described in Section 2.2. The 3.97 µm sample was also measured without a water reference. The water absorption between 800 nm and 1000 nm superimposes the scattering signal of the microspheres in this sample and differs from the Mie calculations. For shorter wavelengths, the scattering is the dominant part of the extinction of the sample and, therefore, the Mie calculations still fit well. The lower graph shows the relative error between the measured data for the 3.97 µm sample and the corresponding Mie calculation. The relative error is less than ±0.7% for the displayed spectral range and the relative mean error is 0.17%.

Figure 11 shows the measurement of polystyrene microspheres with a diameter of 3.917 µm and a standard deviation of 0.03 µm over the full spectral range. A thin cuvette $d_{cuv2}$ = 500 µm and an increased concentration of scatterers were used to minimize water absorption in the NIR, which is the main contribution to the extinction results from the scattering of the sample. The use of demineralized water as a reference leads to good agreement between measurements and calculations using Mie theory. Around the absorption band of water at 1950 nm, the extinction was still too high to be correctly evaluated. The missing data have been interpolated linearly. For the measured data at wavelengths shorter than 300 nm, the intensity of light was not sufficient to be evaluated. The lower graph shows the relative error between the measured and the calculated data using Mie theory. The error over the full spectral range is smaller than ±2.3% and the relative mean error is 0.8%.

## 4. Discussion

A broadband collimated transmission spectroscopy setup was developed and validated. Several improvements were investigated to increase the accuracy of the measurements. It was also shown theoretically that multireflection in cuvettes must be taken into account to correctly evaluate the measurements. Neglecting these effects leads to errors of up to several percent, depending on the sample. Furthermore, the error due to forward scattering was theoretically investigated and can be of large significance, depending on the scattering phase function. For highly forward-scattered samples, errors of up to 35%, even for the optimal optical thickness, can occur. Optimal values for the collimated transmission measurements were identified to be in a range between 25% and 50%. Measurements ranging from UV to IR were presented and evaluated. This gives the possibility to characterize samples in a wide spectral range with high accuracy. The overall improvements and insights presented are useful for all similar applications and systems, especially standard absorption spectroscopy, which should include multireflection evaluation. The system was validated using polystyrene microspheres with two different diameters of 3.91 µm and 2.002 µm. The mean relative error between measurements and calculations is 0.8% over the whole spectrum. For the restricted measurements, carried out first, between 400 nm and 1000 nm, the mean relative error is below 0.2%. To further improve the accuracy of the system, the use of polarization filters to suppress the multiscattered light and a larger angular scanning range to characterize forward-scattered light are options. To realize background scanning, it is already technically possible to scan a defined angular space around the forward direction, but this is not implemented yet. This is particularly relevant for highly scattered samples and will be the subject of further work.

## 5. Conclusions

A system has been developed that has an auto aligning function to compensate for samples that are not exactly plane-parallel. This simplifies the handling of the system and increases the robustness of the measurement. The low NA of the system allows for the measurement of highly forward-scattered samples. The wide spectral measurement range enables a large selection of applications. Theoretical calculations have identified and partially compensated for inherent sources of error in the system. The limitations of the sample properties with respect to optical thickness and scattering phase function were investigated. We point out that the multireflection error analysis is also relevant to standard absorption spectroscopy.

## Figures and Tables

**Figure 1 sensors-24-06993-f001:**
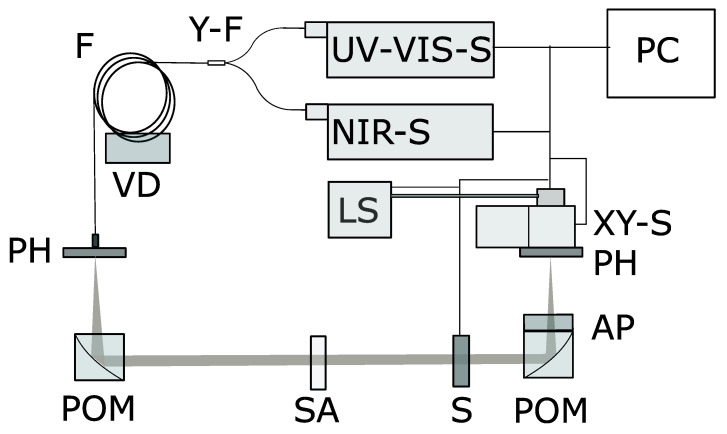
A schematic sketch of the developed measurement setup showing all components as well as the light path. LS: light source; XY-S: xy-stage; PH: pinhole; AP: aperture; POM: parabolic off-axis mirror; S: shutter; SA: sample; F: fiber, VD: vibration device; Y-F: Y fiber, UV-VIS-S: spectrometer for wavelengths 250–1100 nm; NIR-S: spectrometer for wavelengths 900–2150 nm; PC: personal computer used for data processing and device control.

**Figure 2 sensors-24-06993-f002:**
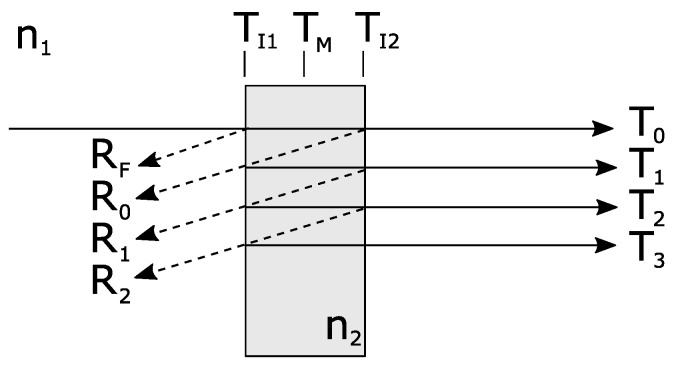
A schematic illustration of different light rays (0 to 4) propagating through a single slab showing different possible propagation paths. The reflection angles are modified unrealistically to obtain a better representation.

**Figure 3 sensors-24-06993-f003:**
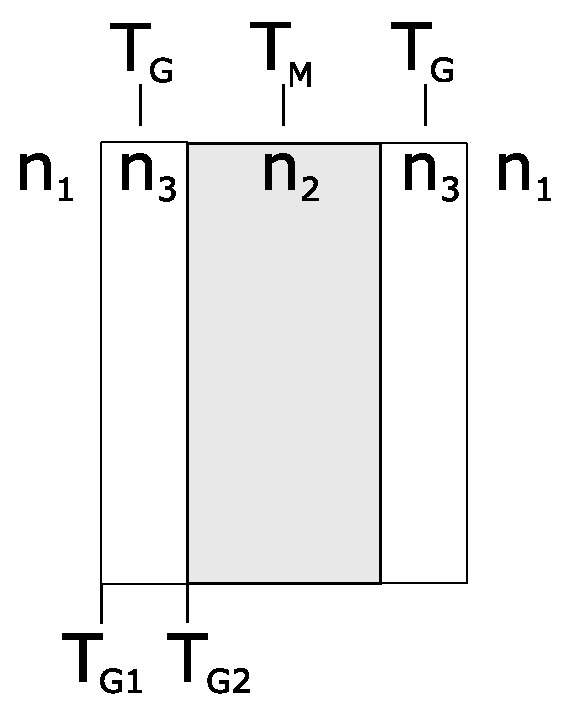
A schematic illustration of a cuvette. $T_{G}$ represents the transmission through the glass slabs, while $T_{M}$ gives the transmission through the medium. $T_{G1}$ and $T_{G2}$ are the transmissions through the interfaces between the surroundings and glass and between the glass and medium, respectively.

**Figure 4 sensors-24-06993-f004:**
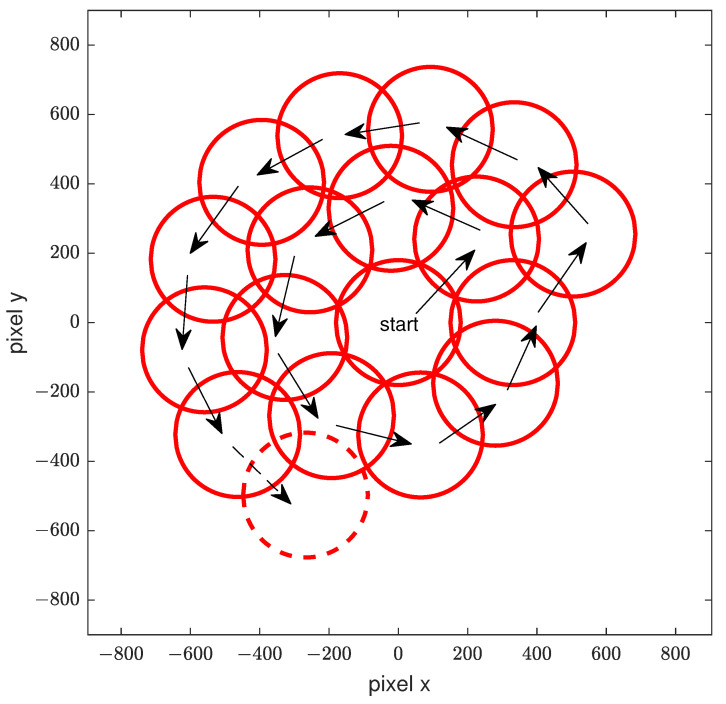
A schematic representation of the movement of the light source in the xy-plane to search for or a sufficiently large measurement signal. The light source is located in the center of the circle which represents the translation area in which a signal will be detected.

**Figure 5 sensors-24-06993-f005:**
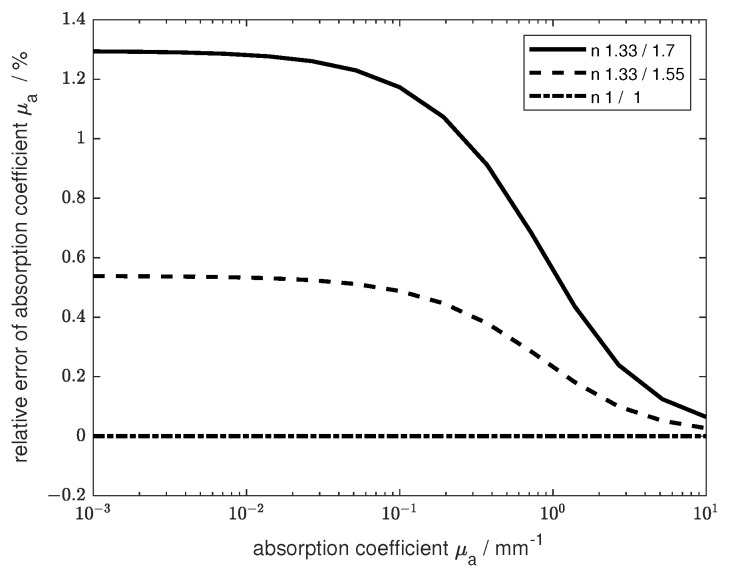
The relative error over the absorption coefficient is given for different refractive indices of the sample and the cuvette. These are coded by line type. The data were calculated using multiple reflection theory.

**Figure 6 sensors-24-06993-f006:**
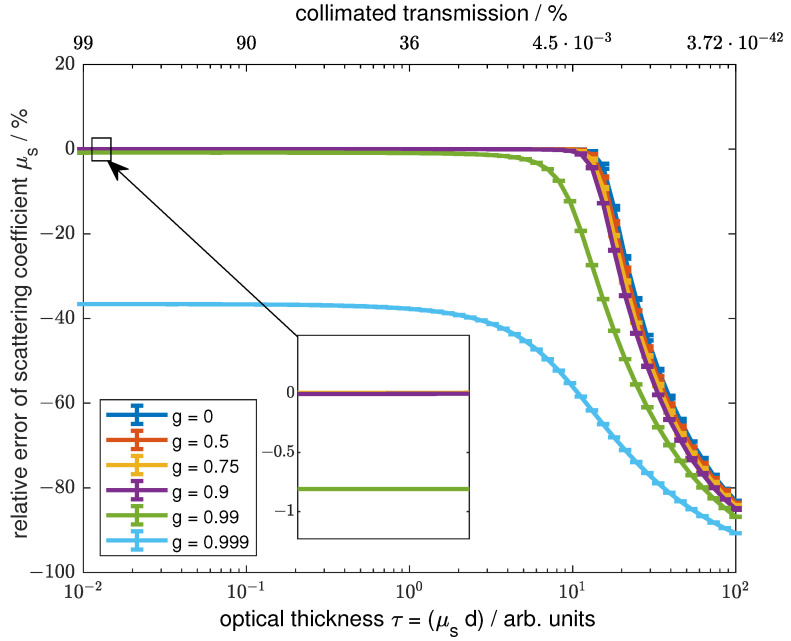
The relative error depending on the anisotropy factor g for a Heneye–Greenstein phase function over the scattering coefficient is shown. The different colors code the used anisotropy factor. The zoom shows a close-up look on the low scattering area for the anisotropy factors < 0.999. The data were generated using Monte Carlo simulations.

**Figure 7 sensors-24-06993-f007:**
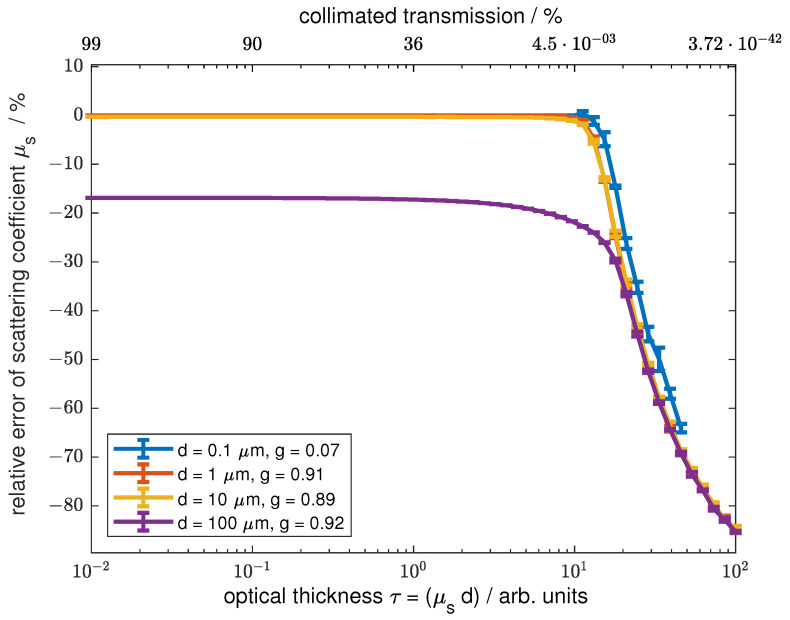
The relative error depending on the phase function calculated for Mie scatterers over the scattering coefficient is shown. The different colors code the used scattering diameter in the Mie calculations and the anisotropy factor calculated for the respective phase function is given in the legend. The data were generated using Monte Carlo simulations.

**Figure 8 sensors-24-06993-f008:**
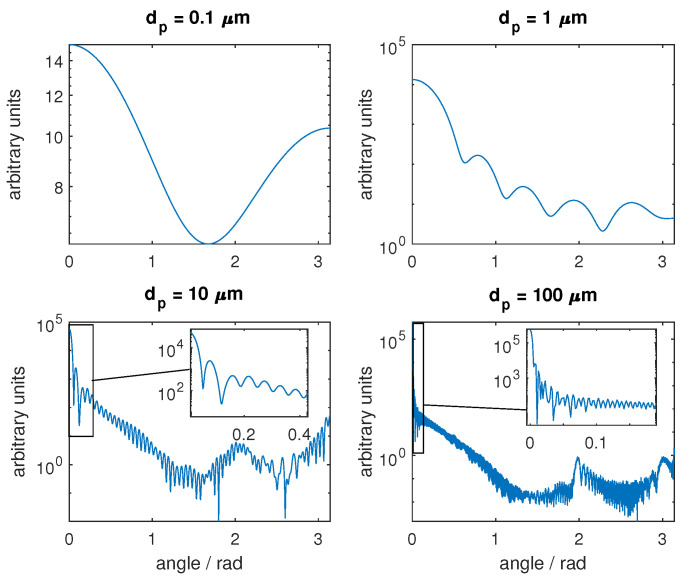
The phase functions for polystyrene spheres with different diameters obtained by Mie’s theory. For $d_{p}$ = 10 µm and $d_{p}$ = 100 µm, the area around forward scattering is presented in a zoomed window. The data were generated using theoretical Mie calculations.

**Figure 9 sensors-24-06993-f009:**
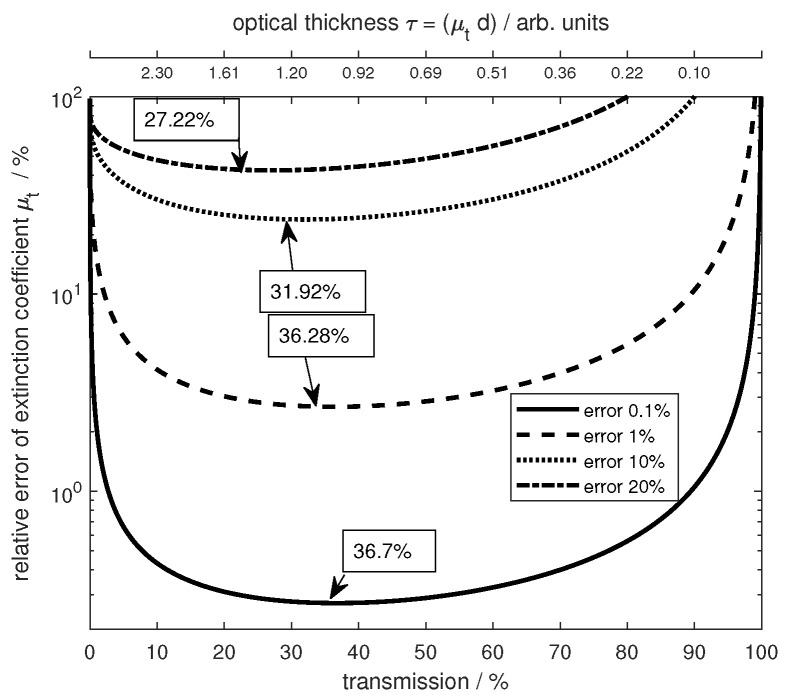
The relative error of the extinction coefficient versus the degree of transmission for various added systematic errors in the transmission signal. Different line styles indicate the errors. The top x axis gives the optical thickness equivalent to the transmission. The boxed numbers give the the transmission value for the smallest error of the respective curve. The data were calculated using the multireflection theory.

**Figure 10 sensors-24-06993-f010:**
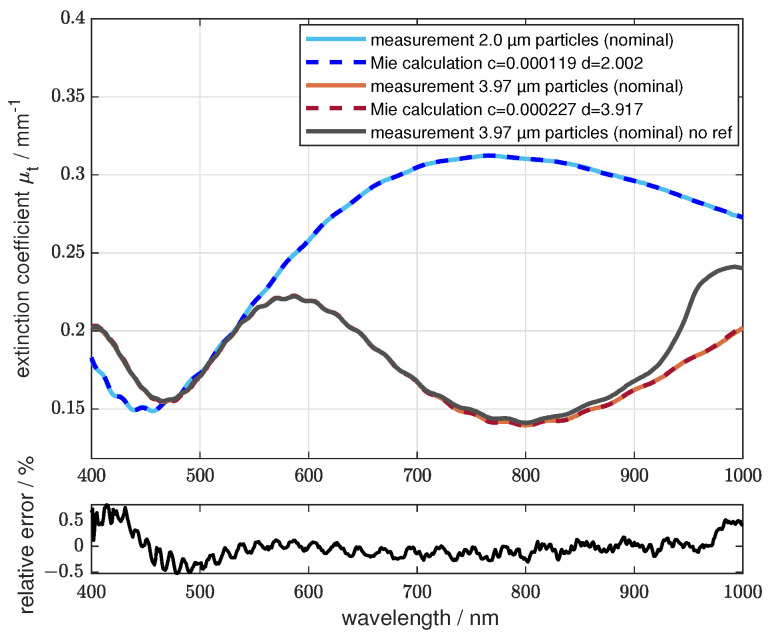
**Upper** Graph: Measurements of the extinction coefficient compared to Mie calculations of the scattering coefficient. Polystyrene spheres with a diameter of 2.002 µm and 3.917 µm were used (nominal 2.0 µm and 3.97 µm). The c value gives the volume concentration of the scatterers. **Lower** Graph: The relative error of the measurement of the 3.917 µm polystyrene spheres with the corresponding Mie calculation over the wavelength.

**Figure 11 sensors-24-06993-f011:**
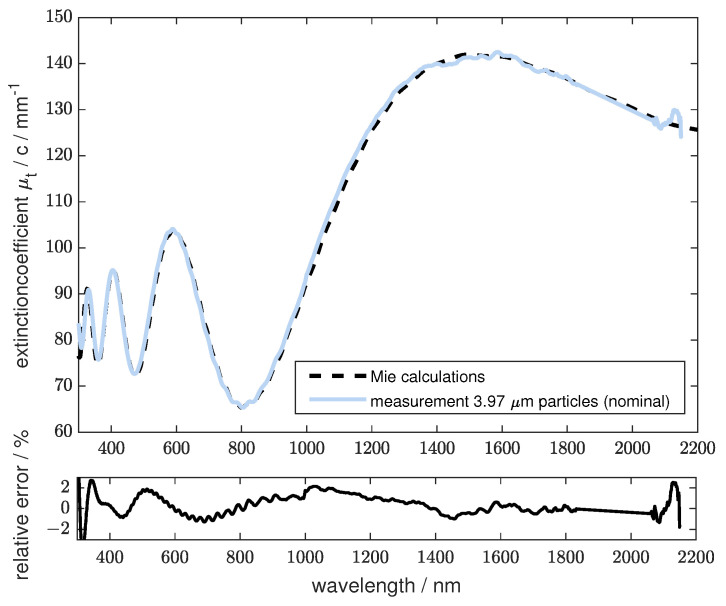
**Upper** Graph: Measurements of the extinction coefficient compared to Mie calculations of the scattering coefficient. Polystyrene spheres with a diameter of 3.917 µm (nominal 3.97 µm) with a standard deviation of 0.03 µm were used. The full possible spectrum that can be acquired is shown. At the water absorption band at 1950 nm, the measured signal could not be evaluated. The data in this region are interpolated linearly. **Lower** Graph: The relative error over the wavelength of the above measurement with the corresponding Mie calculation.

## Data Availability

The data generated during the research presented in this paper are available on request. The raw data as well as detailed information on the setup will be provided by the authors.

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
