# Peer review of "Auto Aligning, Error-Compensated Broadband Collimated Transmission Spectroscopy"

_sensors, 2024, doi:10.3390/s24216993_

Round 1

Reviewer 1 Report

Comments and Suggestions for Authors

The paper is written clearly and thoroughly, the pictures clearly illustrate the presented results. I think the obtained results will represent useful information for future research by the wider scientific community.

Reviewer 2 Report

Comments and Suggestions for Authors

This manuscript presents a novel, self-aligned, broadband, collimated transmission spectroscopy device with error compensation that could be very useful for measurements on various materials such as food and food substitutes, restorative work, e.g. on teeth, as well as for tissue-like phantom applications.

This also shows that the optical transmission is in a range between 25 and 50 %, which is also confirmed by the authors' Monte Carlo simulations, according to which the transmission must be greater than 10 % and less than 90 % to enable reliable and repeatable measurements and to avoid noise and systematic errors. The setup also demonstrates how highly sensitive and selective it can be at very small numerical aperture (NA) and when using conventional glass cuvettes. This research is very useful for the understanding of scattering and absorbtion in various scenarios of transmission spectroscopies with turbid liquid or coloured solid samples, ranging from UV-VIS, NIS and Raman spectroscopy. Some minor details could be worked out in more detail, as indicated below. In general, the manuscript is well written, but here are a few minor comments and suggestions to improve it:

    1. Abstract, line 14, please indicate whether the NA number more clearly for the reader. When I read it first, I thought it was a typo but it seems not the case. Perhaps using x as delimiter might be good to use here between the two numbers otherwise it could lead to confucions. This also could be applied to the resulting NA in line 66 and 70.

    2. Figure 1 is generally well presented but as a suggestion, could the schematics be improved to show the novel setup more appealing i.e. the presentation with the help of three-dimensional illustrations.

    3. Chaption of Fig. 1 at the describtion of the vibrational device is a typo. Please change from VB to VD as you indicated it correctly in the drawings of the figure.

    4. In the section 2.4, starting with “Furthermore, it is technically possible to sample” and the following two sentences are rather suggestions to which do not belong to the M&M section. This preferably could be moved to the discussion or conclusion sections.

    5. I was surprised that the conclusion is missing entirely. I would suggest to inlude it and put a focus on the highlights of the setup which were discovered and some additional words about the results of the Monte Carlo (MC) simulation which comes a bit short here. Otherwise the MC was well described in the M&M section but since it is an extension and in-house development of citation [15], it may be interersting for the reader to understand its key advantages.

    6. Furthermore, it was diffcult to understand which results are obtained by experiments and which are the results of the simulations. Especially, this could be more clearly stated in the presentation of the results of the figures 5 to 11.

    7. In the section 3.1, the standard 10 mm cuvette size was shortly described. However, I missed the more specific describtion in the M&M section. More specifically, the mechanical and optical properties could be described such as of which material, wall thickness etc. the cuvette is made of. Please also indicate the manufacturer of the cuvette used in this research.

    8. Line 201, “The only problem is that the refractive indices must be known”. Does this sentence makes sense here? Perhaps it could be expressed differently.

    9. Figure 5 to 11 seem of poor quality in the PDF version of the manuscript. It looks okay when printed out on paper but please make sure you submit the best possible resolution for the figures. The top axis labeling of figure 6 and 7 is too close to the top values. Please increase the distance of those. Also, I would suggest to use en-dash instead of just minus 1 for the absorption coefficient at the bottom axis labeling.

    10. The headline of 3.4 starts with a typo, line 263.

    11. Another typos are in line 292 and the caption of Figure 11 at the last sentence.

Reviewer 3 Report

Comments and Suggestions for Authors

This manuscript is devoted to development of approach to calculation and measuring the transmission of UV and IR radiation through the media containing the scattering samples. The simulations of transmission using MonteCarlo method  were carried  out  and some errors determined by angular distortions were estimated. Authors developed the experimental setup including two spectrometers (NIR and UV-VIS spectrometers) and experimental measurements of transmission were carried out.  The optimal working parameters of the system were determined. The topic of manuscript is important for scientific and technical groups in areas of spectroscopy and its applications.

There are some points to make the information more clear or to correct some details:

1)      The numbers of references in the text should be separated by commas, if they follow each other.

2)      It is necessary to explain the abbreviations at first mention. There are the abbreviation without explanations (e.g., the abbreviation of "NA" for numerical aperture, possibly (the first mention is in 67th line); the abbreviations of "UV-VIS-S" and "NIR-S" for ultraviolet-visible and near infrared spectrometers (74th -75th lines) (there are the wavelength ranges of these spectrometers further, but the abbreviations were entered by names of spectral ranges) ). The abbreviation of "LS" (59th line) is for laser source, possibly, but not laser only (firstly, the letter "S" in “LS”should be described, and secondly the term of "source" includes the unit based on laser, pumping lamp, control units etc. on the whole). should be explained. The abbreviation of "PC" for personal computer explained in Figure 1 caption should be explained at first mention (83rd line)

3)      The designations of the refraction indices of n1 and n2 of environment and sample respectively in Figure 2 and in formulas (1), (2) require the explanation. In addition the use of same letter "n" for refraction index (despite the presence of numbers) and summation index in (5), (6) is not convenient.

4)      It is necessary to explain all values (I,I0, c, d) in equation describing the Lambert-Beers’s law (187th line). In addition the one letter “d” designates different parameters in the text (e.g.,samples had a thickness of d = 10 mm” (218th line), “for particle sizes of d <= 10 µm”(235th line) “a thin cuvette  of d = 500 µm” (285th-286th lines)). It will be better to add the specific subscripts for each types of values marked of one letter.

5)      Authors write “The refractive indices for the second shown calculation were chosen to fit a standard glass cuvette filled with a water based sample.” (193rd - 194th lines). The addition about type of line (dashed line for water sample) corresponding the in Figure 5 described here is necessary there or in Figure caption.  

6)      It is necessary to explain which designations mt, ms (99th line), ma (Fig. 5) correspond to the absorption or scattering coefficients and to which media. The words about “absorption coefficient…mt” and “scattering coefficient ms“  appeared more further  in 187th and  219th line, respectively, But “extinction coefficient mt” appeared in Figure 9,  The “absorption coefficient ma” appeared suddenly in Figure 5.

7)      It is necessary to enter the designation “g” for anisotropy factor in text (223rd line )  with clarifying it and Figure 6 caption because it appears further (244th line). The explanation of the parameters in this formula (p, q) is needed.

8)       It is necessary to check the expression for transmission rate “t” (254th line), the ratio of intensities determines the transmission, but not their product.

9)      Some notes to Figure 10. This Figure is presented in grey colors, and there are two grey solid lines and two black dashed lines  in this Figure. It is necessary to present tis Figure in colors or to change the types of line for one of these couples of graphs. What is “c” in the legend? Additionally, the value 3.917 is rounded to two significant decimal places after point as 3.92. It is necessary to either round correctly or leave three digits after point as for the other diameter 2.002.

10)  It is necessary to check the text and unify the spelling of terms (e.g., “microspheres” (212th line) “micro spheres” (214th line). There are some errata (e.g., “Experiantal” (263rd line) in title of part 3.4; the word “particle” in legend of Figure 10 is written in German as “partikel”; “The lowe Graph” (292nd line) ),  missing the backspaces  (e.g. “well.The” (281st line), “calculation.The” (282nd line) etc.),  

This manuscript is written clear and describes with details the results of developed approach to calculation and measuring the transmission of UV and IR radiation through the media containing the scattering samples.

The manuscript can be published after minor revisions.
